# Fertility Assessment after Ovarian Transposition in Children and Young Women Treated for a Malignant Tumor

Julie Valduga [1,†], Géraldine Desmules [2,†], Line Claude [3], Pascal Chastagner [1], Valérie Bernier-Chastagner [4], Perrine Marec-Berard [2,*] and Christine Rousset-Jablonski [5,6,*]

[1] Department of Paediatric Haematology and Oncology, University Regional Hospital Center, 54000 Nancy, France; julievalduga@icloud.com (J.V.); p.chastagner@chru-nancy.fr (P.C.)

[2] Department of Pediatric Oncology, Léon Bérard Cancer Center, Institute for Paediatric Haematology and Oncology, 69008 Lyon, France; gdesmules@numericable.fr

[3] Department of Radiation Oncology, Centre Léon Bérard, 69008 Lyon, France; line.claude@lyon.unicancer.fr

[4] Department of Radiation Oncology, University Regional Hospital Center, 54000 Nancy, France; v.bernier@nancy.unicancer.fr

[5] Department of Surgery, Léon Bérard Cancer Center, 69008 Lyon, France

[6] INSERM U1290 RESHAPE, Hôpital Femme Mère Enfant, 69500 Lyon, France

[*] Correspondence: perrine.marec-berard@ihope.fr (P.M.-B.); christine.rousset-jablonski@lyon.unicancer.fr (C.R.-J.)

[†] These authors contributed equally to this work.

**Abstract:** Ovarian transposition (OT) has been proposed as a protective measure against radiation-induced damage to ovarian function and fertility. Despite its historical use, limited research has focused on evaluating endocrine and exocrine ovarian function after OT performed in adolescents and young adults (AYAs) before or during puberty. The purpose of our study was to investigate the fertility, pubertal development, and ovarian function of women with a previous history of OT during childhood, adolescence or young adulthood. In an observational bicentric retrospective study, we included 32 young female cancer patients who underwent OT before the age of 26 between 1990 and 2015 at Lyon Léon Bérard Cancer Center or Nancy University Hospital. The mean age at the time of OT was 15.6 years with a cancer diagnosis at $15 \pm 4.8$ years. Among the 10 women attempting pregnancy post-treatment, 60% achieved successful pregnancies. After a mean follow-up of $9.6 \pm 7$ years, 74% (17 out of 23) of women recovered spontaneous menstrual cycles (seven out of eight evaluable women with OT before or during puberty). Notably, 35% of women who did not attempt pregnancy demonstrated adequate ovarian reserve. Ovarian reserve and function recovery were influenced by the specific chemotherapy received. Importantly, our findings suggest that OT's effectiveness on ovarian activity resumption does not significantly differ when performed before or during puberty compared to pubertal stages. This study contributes valuable insights into the long-term reproductive outcomes of young women undergoing OT, emphasizing its potential efficacy in preserving ovarian function and fertility across different developmental stages.

**Keywords:** fertility; ovary; antineoplastic agents; pregnancy; radiotherapy; AYA; childhood cancers

## 1. Introduction

Approximatively 350,000 cancers are diagnosed in France every year: 1750 cases in patients under 15 years, 800 in those aged 15 to 19 years, and 25,000 in individuals between 20 and 44 years old [1,2]. Advances in patient care and antineoplastic treatments since the 1970s have markedly increased survival rates, with 82% of children, adolescents and young adults (AYAs) in Europe and the United States surviving at least 5 years post-cancer diagnosis, and many achieving long-term survival into adulthood [1,3]. However, despite these positive trends, cancer therapies may impart long-term complications, particularly affecting reproductive functions.

Ovarian dysfunctions are multifaceted, stemming from gonadotoxic chemotherapies, pelvic and/or brain radiotherapy, and pelvic surgery. The gonadal toxicity of certain chemotherapies, particularly alkylating agents and platinum salt derivatives, has long been recognized as a major contributor to fertility issues. Radiotherapy can also impact reproductive function through various mechanisms such as pelvic irradiation (due to the high radiosensitivity of ovarian tissue and the risk of uterine fibrosis or endometrial atrophy) and brain irradiation (hypothalamic–pituitary damage, which can lead to hypogonadotropic hypogonadism). Similarly, pelvic surgery may also be responsible for infertility in cases of oophorectomy, hysterectomy, cystectomy, and salpingectomy.

In compliance with French bioethics laws, fertility preservation (FP) is mandated for all patients facing potentially gonadotoxic treatments [4]. Various methods are now available. Oocyte/embryo freezing (exclusively proposed to post pubertal women) is considered an established FP method [5,6]. Ovarian tissue freezing followed by transplantation offers promising pregnancy rates. Use of gonadotropin-releasing hormone (GnRH) agonists during chemotherapy is still debated as there is conflicting evidence on GnRHa and other means of ovarian suppression efficiency for FP. Certain techniques, like in vitro maturation, and oophoropexy, also called ovarian transposition (OT), are still considered experimental methods. Notably, OT as the pioneering surgical procedure for preserving ovarian function in the context of abdominal/pelvic radiotherapy (RT) [7], involves repositioning one or both ovary(ies) outside the radiation field. Despite its historical significance, the potential efficacy of OT for FP in young patients remains inadequately elucidated.

Hence, this study aims to explore fertility outcomes in cancer patients under 26 years who underwent OT, offering insights into the viability of this surgical technique during therapeutic interventions. Additionally, we investigated its impact on pubertal development and ovarian function post intervention. By addressing these critical gaps in knowledge, we aspire to contribute valuable data to the evolving landscape of fertility preservation strategies for young cancer patients.

## 2. Materials and Methods

### 2.1. Patients and Inclusion Criteria—Data Collection

This study included patients who had been diagnosed with malignant tumors and subsequently underwent OT before the age of 26, at either the Léon Bérard Cancer Center (CLB) in Lyon or the University Regional Hospital Center (CHRU) in Nancy, spanning the period from 1990 to 2015. Women who had died at the time of the study (n = 6) were excluded from the analysis. Women younger than 16 years (n = 21) at the time of the study (at the time of the hormonal evaluation), were also excluded, as they were too young to be able to evaluate their procreative project during the follow-up. We retrospectively analyzed the institutional electronic medical records.

### 2.2. Cancer Treatment and Disease Follow-Up

Data on the histological diagnosis, stage of the disease, and specifics of the treatment administered (protocol, drugs, cumulative doses) were meticulously collected. In case of RT, target volume, RT total dose, fractionation schedule, and radiotherapy techniques were also gathered. Dosimetry parameters, both average and maximum doses, were retrospectively reviewed on dose-volume histograms at the local pelvic level (ovaries, uterus) and, in the case of craniospinal RT, at the pituitary level.

### 2.3. Fertility Preservation

For OT, the surgical reports were considered to determine the surgical approach (laparotomy or laparoscopy), the ovary(ies) subjected to transposition, and the designated re-location site. In case of unilateral transposition, the contralateral ovary outcome was specified. The occurrence of operative complications related to the OT procedure was also systematically assessed. For each patient, we also noted whether additional procedures

for FP were used: cryopreservation of ovarian tissue, ovarian stimulation with oocyte vitrification or embryo freezing, and GnRH agonists' use.

*2.4. Assessment of Fertility and Ovarian Function*

Information regarding puberty and menstrual cycles was systematically collected at three key time points: at the time of cancer diagnosis (when OT was performed), at the conclusion of cancer treatment, and at the time of this study. This comprehensive dataset included details on the nature of puberty onset (spontaneous or pharmacologically induced) and age at the onset of puberty and menarche. All participants had a gynecological consultation as part of their routine care. If the patient was unavailable for an in-person consultation, a reproductible interview was conducted by phone, ensuring a standardized data collection approach. Information regarding contraception, attempts at pregnancy, recourse to assisted reproductive technology, and the occurrence of spontaneous pregnancy were evaluated. In case of pregnancy, the time to conceive and the pregnancy outcome were also studied. Ovarian function was rigorously assessed through the analysis of menstrual cycles' regularity along with blood levels of estradiol, follicle-stimulating hormone (FSH), and luteinizing hormone (LH). Premature ovarian failure was defined as the occurrence of amenorrhea > 4 months before 40 years old, associated with an FSH level > 30 IU/L and a decreased level of estradiol on two successive hormonal tests. The ovarian reserve was estimated using anti-Müllerian hormone (AMH). Low AMH was defined as a level < 5 pmol/L. An antral follicle count, determined by ultrasound, was also conducted, with a low follicle count defined as fewer than 5 in one ovary or 10 in both ovaries.

*2.5. Statistical Analysis*

Continuous quantitative variables are presented as means with standard deviation or 95% confidence interval. Medians were used when the degree of dispersion was substantial, with the minimum and maximum range (E = [minimal value–maximal value]). Qualitative variables are presented as frequencies and proportions. The analyses were performed using SAS v9.4 software (SAS Institute).

*2.6. Ethical Approval and Consent*

This study was approved by the Protection of Persons and Property Committee of the Léon Bérard Cancer Center (CLB) in Lyon and was registered by the National Commission for Data Protection and Liberties, France.

**3. Results**

*3.1. Population*

Between 1990 and 2015, a total of 59 OTs were performed in patients under the age of 26 years (Figure 1), all of whom were undergoing treatment for various malignant tumors, as show in Figure 2. Among these procedures, 53 were conducted at the Léon Bérard Cancer Center (CLB) and 6 at the University Regional Hospital Center (CHRU). Twenty-seven patients were excluded from the analysis, comprising 6 individuals who succumbed to their condition and 21 who were younger than 16 years at the time of the study. After applying inclusion and exclusion criteria, 32 patients were included in the study, with an average age of 24.4 years old at the time of assessment. The mean age at the cancer diagnosis was $15 \pm 4.8$ years of age. The average follow-up period after the cancer treatment was $9.6 \pm 7$ years. Notably, 12 patients (38%) experienced a relapse during the course of follow-up.

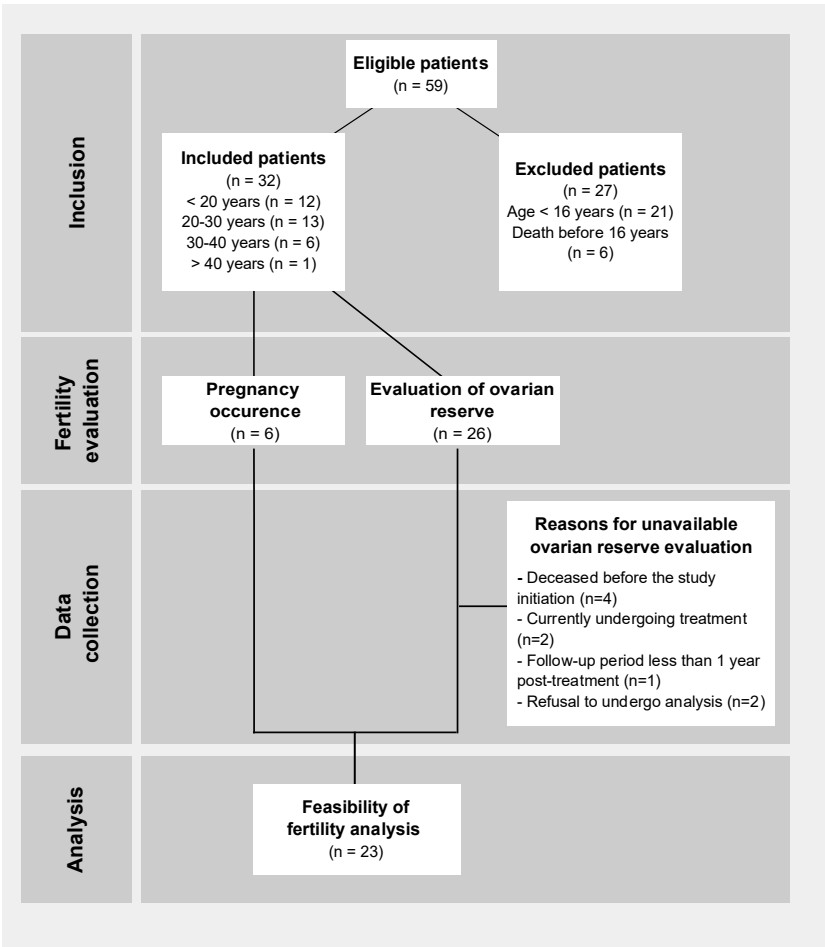

**Figure 1.** Flowchart of analyzed patients.

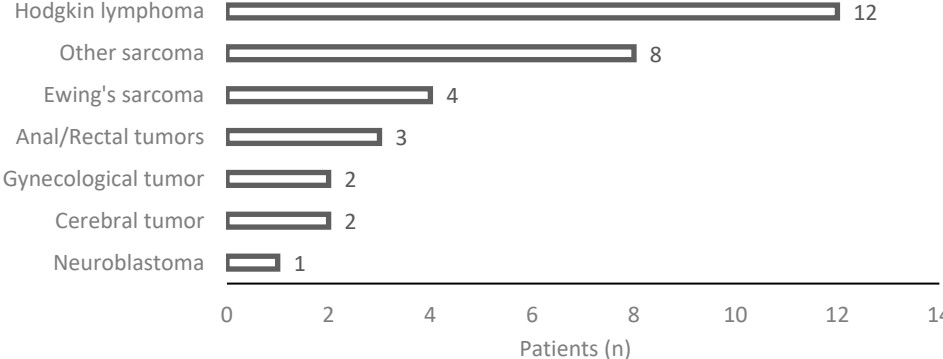

**Figure 2.** Details on histological diagnosis.

### 3.2. Cancer Treatment

Cancer treatments involved a combination of chemotherapy, and/or radiotherapy, and/or surgery, as illustrated in Figure 3. The specific details of the treatments, including cumulative chemotherapy doses and radiotherapy doses, are summarized in Table 1. Among the 32 enrolled patients, approximately 56% (n = 18) underwent surgery. The majority, 81% (n = 26), received chemotherapy combined with RT, while a smaller proportion, 12.5% (n = 4), did not undergo any chemotherapy. Pelvic RT was administered to 28 women (87.5%), craniospinal irradiation to two patients, and two individuals did not ultimately receive RT despite the initial treatment plan. Of those receiving RT, 21 patients underwent conformal RT, five were treated with intensity-modulated RT, three with brachytherapy, and

one with hypofractioned stereotaxic radiation therapy. Most patients received associated chemotherapy and the mean dose to the ovaries varied individually. The diverse treatment modalities underscore the heterogeneity within the cohort, underlying the complexity of therapeutic strategies employed in this population.

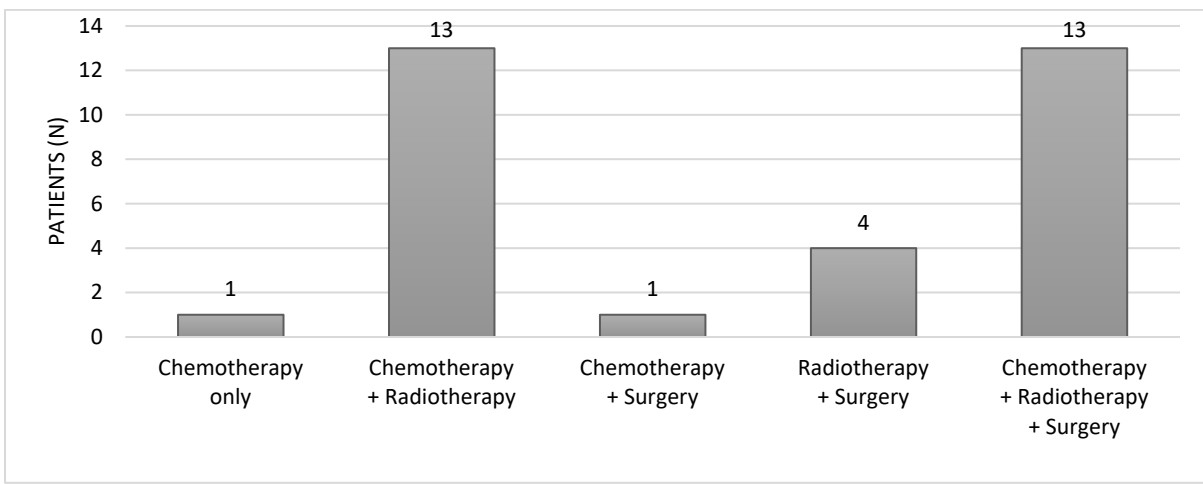

**Figure 3.** Cancer treatment approaches.

**Table 1.** Detailed cancer treatments.

| Treatment | n (%) | Mean | SD | 95% CI | Median | Min [1] | Max [1] |
|---|---|---|---|---|---|---|---|
| **Chemotherapy (cumulated doses)** | 28 (88%) | | | | | | |
| Busulfan (mg/m$^2$) | 3 (9%) | 429 | 321 | [66–793] | 247 | 241 | 800 |
| Melphalan (mg/m$^2$) | 5 (16%) | 716 | 498 | [152–1280] | 1000 | 140 | 1007 |
| Ifosfamide (g/m$^2$) | 15 (47%) | 41 | 24 | [29–53] | 45 | 7 | 95 |
| Cyclophosphamide (mg/m$^2$) | 19 (59%) | 4721 | 3249 | [3220–6222] | 3946 | 554 | 11,863 |
| Procarbazine (mg/m$^2$) | 9 (28%) | 6315 | 3674 | [3769–8860] | 8959 | 2094 | 9028 |
| Dacarbazine (mg/m$^2$) | 6 (19%) | 2968 | 1917 | [1434–4501] | 2849 | 747 | 5425 |
| **Radiotherapy** | 30 (94%) | | | | | | |
| Total dose (planning target volume) (Gy) | - | 36 | 16.5 | [30.3–41.7] | 36 | 19.8 | 60 |
| <20 Gy | 1 (3%) | - | - | - | - | - | - |
| >20 Gy | 29 (97%) | - | - | - | - | - | - |
| Number of fractions | - | 20 | 9 | [17–23] | 20 | 5 | 36 |
| Duration | - | 33 | 15 | [28–38] | 36 | 11 | 65 |
| Mean dose [2] to ovaries (Gy) | - | 3.3 | 7.4 | [0.7–5.8] | 1 | 0 | 36 |
| Maximal dose [3] to ovaries (Gy) | - | 3.6 | 7.6 | [0.7–6.5] | 2 | 0 | 36 |
| <3 Gy | 12 (38%) | - | - | - | - | - | - |
| 3–10 Gy | 7 (22%) | - | - | - | - | - | - |
| 10–20 Gy | 5 (16%) | - | - | - | - | - | - |
| >20 Gy | 8 (25%) | - | - | - | - | - | - |
| Mean dose [2] to uterus (Gy) | - | 14.3 | 14.2 | [6.5–22.0] | 20 | 0 | 45 |
| Maximal dose [3] to uterus (Gy) | - | 20.9 | 17.4 | [13.1–28.7] | 20 | 0 | 54 |

Abbreviations: CI, confidence interval, SD, standard deviation. [1] Referred to the range of values within patients. [2] "Mean dose" refers to mean dose received by the organ for each patient. [3] "Maximal dose" refers to the maximum dose received by the organ.

### 3.2.1. Fertility Preservation

OT was conducted at an average of 7.6 ± 4.3 months after cancer diagnosis, with a mean delay of 1.2 months before the initiation of RT. The majority of OT procedures were performed laparoscopically in 84% (n = 27) of the cases, while laparotomy was employed for five patients (16%), exclusively during cancer surgery. Both ovaries were transposed

in 65.6% of the cases (n = 21) (Figure 4). All women underwent OT after sectioning of the utero-ovarian ligament and pedicularization of the adnexa on the infundibulopelvic vessels to preserve vascularization. The OT location depended on the irradiation fields evaluated by RT treatment planning. The ovary(ies) were moved laterally in the paracolic gutter for 30 patients (~94%) and medially in a retro-uterine position for two patients (6.25%). Six patients (18.7%) experienced complications related to surgery, as shown in Table 2. Notably, one patient, with a time lapse of 3.1 months between OT and RT, required re-operation during RT due to secondary migration of a transposed ovary.

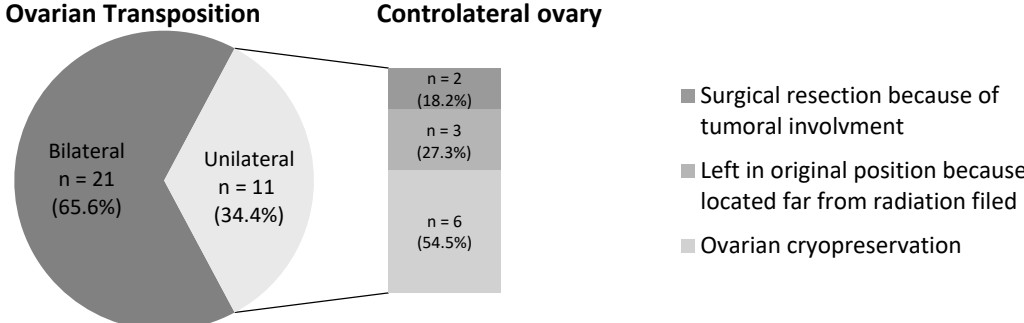

**Figure 4.** Characteristics of ovarian transposition and management of the controlateral ovary in case of unilateral ovarian transposition.

**Table 2.** Complications related to the ovarian transposition and their consequences.

| Timing | Complications | Grading | Consequences |
|---|---|---|---|
| **Per-operative** | Hematoma retro-peritoneal | Grade 3 | Per-operative evacuation |
| | Peritoneal adhesions | Grade 3 | Conversion to laparotomy |
| | Ischemia of fallopian tube | Grade 3 | Homolateral salpingectomy |
| **Post-operative** | Bladder injury with dysuria | Grade 3 | Bladder suturing |
| | Chronic abdominal pain | Grade 2–3 | Reduction of quality of life |

Regarding fertility preservation, OT was the sole method employed in 19 patients (~59%). Additional strategies included OT combined with GnRH agonists in 16% of cases, OT paired with controlateral ovarian cryopreservation in 9%, and a combination of OT, GnRH agonists and ovarian cryopreservation in 16%. Notably, none of the patients had oocyte or embryo cryopreservation. The majority of women underwent bilateral OT.

3.2.2. Pubertal Development and Ovarian Function (Menstrual Cycles)

The mean age at the time of OT was 15.6 years [14.0–17.2] (range 4.7–25.8). Twenty-seven patients (~84%) had already exhibited pubertal symptoms, and 23 (~72%) had menstrual cycles. According to the pubertal status at the time of OT, patients were classified into three subgroups: prepubescent patients at time of OT, puberty in progress but still amenorrheic at time of OT, and pubescent with menstrual cycles at time of OT (Figure 5). Among the five prepubescent patients, two women who underwent unilateral OT with contralateral oophorectomy experienced temporary primary amenorrhea but eventually had spontaneous menarche. Thus, all five patients who underwent prepubertal OT had a spontaneous menarche, and all four patients who were evaluable at the time of this study were still menstruating naturally. These patients received pelvic conformal 3D RT with ovar-

ian dose < 3 Gy in combination with high-dosage chemotherapy and autologous stem cell transplantation. Among the four patients who started puberty without menstruation before OT, two began menstruating one month after the end of the cancer treatment. However, one of them developed primary amenorrhea after pelvic conformal 3D radiotherapy with ovarian irradiation of 12 Gy and high-doses alkylating chemotherapy, while the other presented secondary amenorrhea after a relapse treated by high-dosage alkylating chemotherapy and allogeneic stem-cell transplantation. Altogether, seven of the eight evaluable patients with OT before or during puberty had recovered natural cycles at the time of the study. By considering the 23 pubescent patients with spontaneous menstruation, three continued to menstruate during treatment and 20 had a temporary amenorrhea (after pelvic RT > 20 Gy). Among them, 45% (n = 9/20) recovered spontaneous menstrual cycles after 5 ± 2 months and 55% (n = 11/20) had secondary amenorrhea, spontaneously resolving for only one patient. To note, among them, four received high-dose alkylating chemotherapy, and three of them received irradiation with a uterine dosimetry greater than 25 Gy potentially responsible for this amenorrhea. Lastly, 15 patients (~47%) exhibited primary or secondary amenorrhea. These patients had been more often exposed to melphalan (33% vs. 0%, *p* = 0.01), but also to busulfan (20% vs. 0%), cyclophosphamide (73% vs. 47%) and ifosfamide (60% vs. 35%), although these results are not statistically significant. No significant differences were noticed when considering the degree of pelvic, ovarian, or uterine RT. Three patients with primary amenorrhea finally had late spontaneous menarche. After the exclusion of deceased women (n = 4), women still under cancer treatment (n = 2) or those using amenorrhea-inducing treatment (tamoxifen, levonorgestrel IUD, progestin-only pill) (n = 3), the rate of spontaneous cycle recovery was 74% (n = 17/23). Long-term amenorrhea needing hormonal treatment to induce menstruation appeared to be less frequent when patients had not yet been menstruating at the time of OT (1/8 = 12.5% vs. 5/17 = 29.4%).

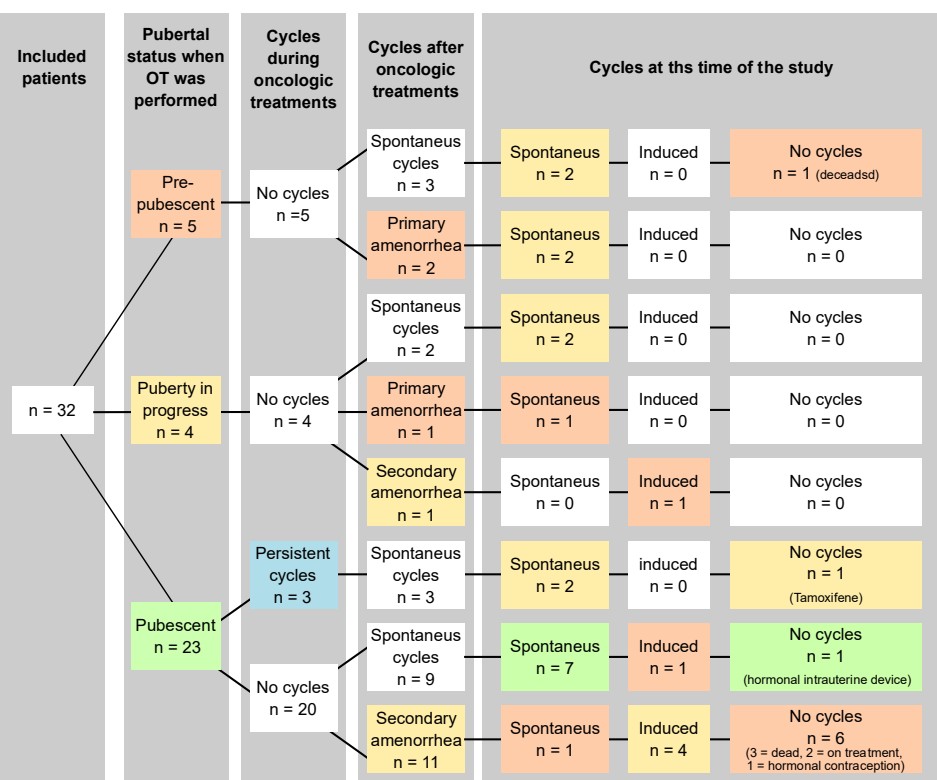

**Figure 5.** Pubertal status at the time of OT, and evolution of pubertal status and ovarian function (menstrual cycles) during treatment, after treatment and at the time of this study.

Most prepubescent or pubescent but amenorrheic patients eventually started puberty and had spontaneous menstruations. Among pubescent patients, most women sponta-

neously recovered ovarian function. The administration of gonadotoxic chemotherapy was associated with the risk of primary or secondary amenorrhea.

Hormonal assessment was performed in all women still alive with complete remission greater than 1 year, consistent with the recovery of ovarian activity. Hormonal blood tests were conducted in 20 patients: 20% were diagnosed with premature ovarian failure (POF), and 10% with hypogonadism hypogonadotropic secondary to cranial RT. Ovarian reserve was analyzed in 17 women, thus revealing nearly undetectable AMH levels (<1 pmol/L) for eight of them (47%) and a decreased AMH level for 18% (n = 3). An antral follicle count was performed in 14 patients and was reduced in nine of them (64%). Thus, despite ovarian function recovery (menstruations), a large proportion of women presented reduced ovarian reserve.

### 3.2.3. Fertility

Ten patients (~31%) within the cohort actively pursued pregnancy, with six achieving success, while four faced primary infertility (Figure 6). Only three patients underwent a second surgery to relocate the transposed ovary(ies) to their original position(s), with a median duration of 7.5 years after cancer treatment. All six pregnancies were spontaneous, occurring at a median of 10.4 years after the end of cancer treatment (E = [0.9–16.5]). For these patients, OT was performed at a median age of 19.4 years (E = [11.6–22.0]), and none underwent ovarian repositioning. The median time to achieve pregnancy was 6 months after the first attempt (E = [1–24]). Only two of them experienced a spontaneous miscarriage. Among the four patients diagnosed with primary infertility, they were younger at the time of the OT compared to those who could have spontaneously become pregnant (median age: 15 vs. 19.5). Two of them were prepubescent when OT was performed. These patients were more frequently exposed to cyclophosphamide (75% vs. 25%) and to higher radiation doses to pelvis (50 Gy vs. 14 Gy), ovaries (14 Gy vs. 1 Gy) and/or uterus (26 Gy vs. 4 Gy). In the whole cohort, approximately 69% (n = 22/32) did not attempt to become pregnant.

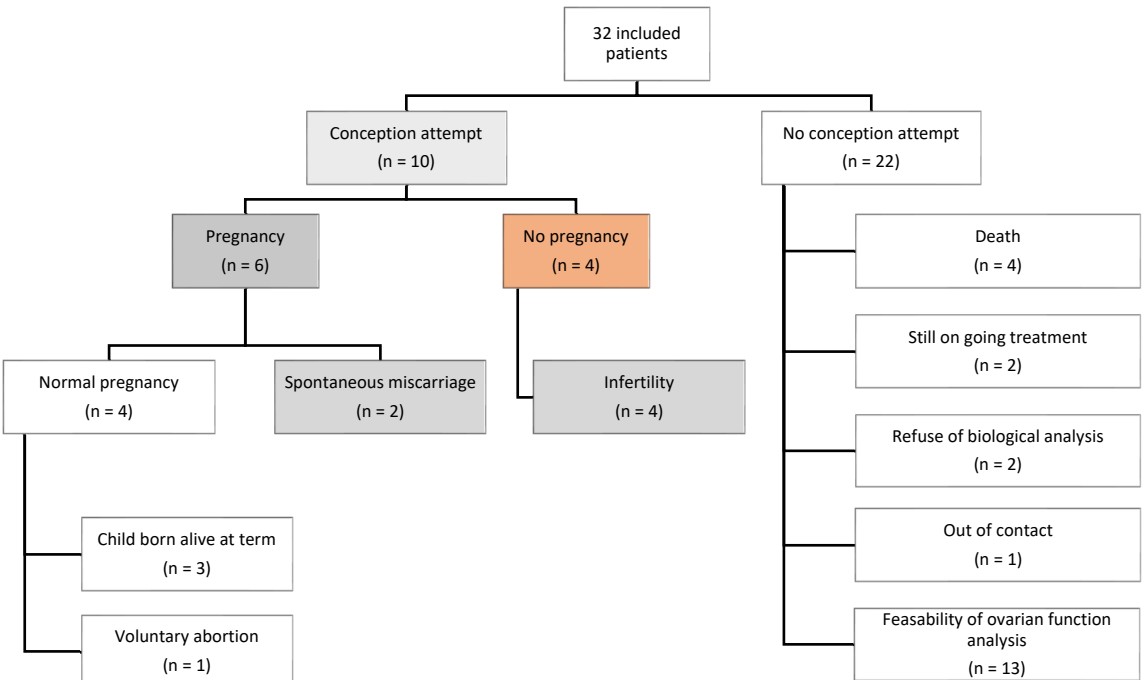

**Figure 6.** Fertility: pregnancy attempts and achievements.

In total, 6 of the 10 women who tried to become pregnant succeeded without ovarian repositioning. The infertile women were more likely to be younger at the time of OT, receive associated gonadotoxic chemotherapy, and be exposed to higher radiation doses.

## 4. Discussion

In this cohort of 32 women undergoing OT before 26 years old, a substantial proportion demonstrated successful pregnancy attempts (n = 6/10) without ovarian repositioning. Similarly, most of the patients (74%) experienced spontaneous or recovered menstrual cycles even when OT had been performed before or during puberty (7 on 9).

However, challenges and uncertainties surround the long-term implication of OT, especially in prepubertal and peripubertal populations. Impairment of reproductive function commonly occurs in childhood cancer survivors. The most gonadotoxic drugs are alkylating agents [8]. RT entails a significant risk of infertility when the ovaries are included in the RT field in a dose-dependent manner [9–12].

OT is employed to reduce the risk of POF [7]. Our findings align with those of Mossa et al. [13], demonstrating a high rate (74% of women recovered spontaneous cycles) of preserved ovarian function after OT. However, despite ovarian function recovery (menstruations), a large proportion of women presented reduced ovarian reserve. This result is difficult to interpret as some patients also received chemotherapy. Gonadotoxic chemotherapy use was associated with an increased risk of primary or secondary amenorrhea. Despite these encouraging outcomes, uncertainties persist regarding the consequences of OT on ovarian function and pregnancy potential, especially when surgery is performed at the pre- or peri-pubertal stages [10,14]. We described that seven out of eight evaluable prepubertal patients finally recovered ovarian function after the completion of their cancer treatment.

The susceptibility to treatment-related infertilities correlates with age and pubertal stage at the time of the treatment [15], with lower gonadotoxicity of chemotherapies and RT in prepubertal females [11,12,16]. While the infertility rate in our cohort was 40% (4/10), the pubertal status at the time of OT did not seem to have an impact. Infertile women were more likely to have received associated gonadotoxic chemotherapy and higher pelvic radiation doses. However, the interpretation of these results is not straightforward due to the small sample size and the majority of patients being younger than 30 years, limiting attempts at pregnancy at the time of this study.

The existing literature, describing treated patients undergoing bilateral OT at a median age of 13 years (range 9–22) [17] and reporting 14 pregnancies with 12 live births, has speculated on the ability of OT to preserve OF and enable future pregnancies. Pubertal status at the time of OT was not described, and thus the efficiency of OT specifically in prepubertal girls was also not described. Notably, only two pregnancies after OT at the prepubertal stage have been reported in the literature [18]. Additionally, two pregnancies occurred in women who underwent an OT at the peripubertal stage. Our study reports 3/6 pregnancies as eutocic, with a miscarriage rate of 33%. The low number of pregnancies in our study is likely attributed to the young age of the included patients, with 69% not actively seeking pregnancy at the time of this study.

Relocation of transposed ovaries to their original positions may not be necessary for fertility restoration, as reported in other studies [19,20]. However, if a medically assisted procreation procedure is contemplated, relocation becomes necessary to facilitate oocyte retrieval [20,21]. In our cohort, a second surgery to relocate ovaries was performed in 9%, while ovaries remained transposed in the six women with spontaneous pregnancies.

Infertility in cancer survivors arises from different factors, complicating the specific estimation of OT benefits. Indeed, infertility could be linked to gonadotoxicity and/or uterine damage in the case of pelvic irradiation [22]. Female fertility can be compromised despite maintenance or resumption of menses. OF and fertility can also be negatively affected by ovarian ischemia after OT [23]. Moreover, the ovary(ies) can be exposed to substantial irradiation despite OT.

Several studies underscored that ovarian preservation is advantageous with a lateral transposition, but the choice depends on a case-to-case basis and the radiation fields [24–26]. The most frequent locations for OT are the paracolic gutters. In our study, bilateral OT to the paracolic gutter was performed in 94% of the patients. Transposition to this level can be readily achieved without separation of the fallopian tubes from their uterine origin,

thereby allowing for spontaneous conception. The most frequent surgical complications are fallopian tube ischemia, ovarian cyst formation, chronic pain, or migration of the ovaries back to their original positions [27]. Considering the risk of migration, the time between OT and the beginning of radiotherapy should be as short as possible. One patient in our series needed to be reoperated during RT due to secondary migration of a transposed ovary in the context of a time lapse between OT and RT of 3.1 months. To avoid the risk of secondary migration, the transposed ovary can be sutured at the peritoneum.

Despite these surgical challenges, the overall incidence of complications was relatively low. Notably, only one patient reported chronic pain and another one experienced perioperative ischemia culminating with a salpingectomy. Intriguingly, this patient became pregnant 12 years after completing her cancer treatment. Remarkably, none of the patients developed ovarian cyst or tumor.

Ovarian suppression through GnRH agonists during chemotherapy is not universally considered to be a method of FP [28,29]. Embryo/oocyte cryopreservation are standard strategies but only concern pubertal patients and may be limited by the need for endovaginal procedures and the impossibility to delay starting the cancer treatment in some cases. In our series, none of the patients underwent ovarian stimulation. Recent studies highlighted that 38% women were able to conceive after ovarian tissue transplantation following gonadal tissue cryopreservation in the post-pubertal period, and that 26% of them had a live birth [30,31]. These results suggest that cryopreservation may become a concrete option. Nevertheless, this procedure remains unproven for ovarian tissue harvested at a prepubertal or pubertal age [18,32]. In this context, OT hence assumes an important role in FP at a young age.

Our study has some limitations. We had no control population without OT, which prevented us from conducting statistical analyses and may limit the interpretation of OT efficiency. OT was performed when the irradiation dose on one or both ovaries was judged to be at high risk of ovarian failure. Thus, despite the absence of a control population without OT, it is estimated that the rate of ovarian failure without transposition would have been very high. Additionally, some patients received chemotherapy, which may have influenced our results, as gonadotoxic chemotherapy could have been responsible for the ovarian reserve damage observed in these patients.

Our results highlight the potential issues associated with the consequences of OT when performed in young female patients, including prepubertal females. However, the small number of patients included in this retrospective study and the heterogeneity of the administered cancer treatments suggest that additional prospective studies, focused on prepubertal populations, are recommended to determine the efficiency of this surgery. These findings suggest the importance of vigilant monitoring and preventive measures during OT procedures, emphasizing the need for a short time interval between OT and the initiation of RT to minimize the risk of secondary migration. While complications were noted, the overall success of the procedure, as evidenced by successful pregnancies and absence of ovarian pathologies, supports the utility of OT as a fertility preservation strategy in this context.

## 5. Conclusions

OT performed in women younger than 26 was associated with a high proportion of resumption of ovarian activity, whether the transposition was performed before or after puberty. Among women who tried to become pregnant, a substantial proportion achieved successful pregnancy attempts without ovarian repositioning. OT presents as a promising approach for preserving both OF and fertility, especially when associated with gonadal tissue cryopreservation or oocyte vitrification, particularly in case of a gonadotoxic chemotherapy. To date, the exploration of OT, specifically in pre- or peri-pubertal children anticipated to undergo pelvic RT, remains inadequately investigated. Hence, further scientific studies are required to meticulously assess fertility outcomes in young women subjected to OT at the pre/peripubertal stage. These additional investigations will contribute to a more nuanced

understanding of the effectiveness of OT as a fertility preservation strategy, particularly within the distinctive context of early-stage cancer treatment.

**Author Contributions:** Conceptualization, C.R.-J., P.M.-B. and L.C.; methodology, C.R.-J., P.M.-B. and L.C.; formal analysis, J.V. and G.D.; investigation, C.R.-J. and J.V.; resources, C.R.-J., P.M.-B. and L.C.; writing—original draft preparation, J.V. and G.D.; writing—review and editing, C.R.-J., P.M.-B., L.C., P.C. and V.B.-C.; supervision, C.R.-J., P.M.-B. and L.C. All authors have read and agreed to the published version of the manuscript.

**Funding:** This research received no external funding.

**Institutional Review Board Statement:** This study was conducted in accordance with the Declaration of Helsinki. Our study was declared to our Institutional Review Board (Protection of Persons and Property Committee of the Léon Bérard Cancer Center (CLB) in Lyon). According to the French Legislation (Loi n° 2004-806 du 9 août 2004 and its subsequent amendments), no IRB approval had to be provided for this retrospective study.

**Informed Consent Statement:** Informed consent was obtained from all subjects involved in this study.

**Data Availability Statement:** The authors confirm that the data supporting the findings of this study are available on demand at christine.rousset-jablonski@lyon.unicancer.fr.

**Acknowledgments:** We thank Erika Vacchelli for her assistance in writing, editing, and critically reviewing the manuscript. We gratefully acknowledge the clinicians who referred patients for the endocrinological analyses.

**Conflicts of Interest:** The authors declare no conflicts of interest.

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
