# Peer review of "Fertility Assessment after Ovarian Transposition in Children and Young Women Treated for a Malignant Tumor"

_curroncol, doi:10.3390/curroncol31060240_

Round 1

Reviewer 1 Report

Comments and Suggestions for Authors

The authors do not compare patients with and without ovarian transposition in this study. Therefore, it remains unclear whether ovarian transposition is effective. This represents a significant limitation of this method. However, the study does address important topics for cancer patients.

Abstract

â‘     You should include the purpose of this study in the abstract.

â‘¡    You state that our findings suggest ovarian transposition's (OT) effectiveness does not significantly differ when performed before or during puberty compared to during pubertal stages. However, based on the methods described in the abstract, it seems these findings could not have been obtained, and thus this should be addressed.

â‘¢    How many patients attempted pregnancy post-treatment?

Material and Methods

The authors describe the same information in both the Materials and Methods and Results sections, which makes the manuscript redundant. This should be improved. For example, 'data collection' is mentioned in the Materials and Methods, and 'population' is discussed in the Results, yet these topics concern the same data. It would be more appropriate to consolidate this information under Materials and Methods.

Additionally, more details should be included about the procedure for ovarian transposition, specifically how it is performed.

Note: There is a spelling mistake in Line 118; it should be 'IU/L', not 'UI/L'.

The purpose of ovarian transposition is to protect the ovaries from radiation therapy (RT). However, this study includes patients who did not receive pelvic RT, which may not be appropriate for evaluating the procedure's effectiveness. It is suggested to exclude patients who did not receive pelvic RT from the study population. In Line 156, you mention that pelvic RT was administered to 28 women.

Table 1

I cannot understand this table, especially the section on radiotherapy. Mean dose to ovaries, Maximal dose to ovaries, Mean dose to uterus, and Maximal dose to uterus is included in Table as Min and Max.

See Line194

You excluded patients under 16 years of age; however, you classified patients into three subgroups. You should reconcile both the inclusion and exclusion criteria. 

Discussion and Conclusion

From line 334 to 353, you should add the results and conclusions obtained from this study. Although the authors highlight ovarian transposition (OT) in prepubertal females, this study excludes patients younger than 16 years old.

Reviewer 2 Report

Comments and Suggestions for Authors

comment 1

In introduction, add more about ovarian preservation methods apart from ovarian transposition (eg: medical treatment with GnRHa) since ovarian cryopreservation appeared in page 6 and IUD containing hormone and OCP appeared in page 7

comment 2 

spelling of voluntary in Fig: 6

Round 2

Reviewer 1 Report

Comments and Suggestions for Authors

The author addressed to my comments. However, this manuscript has several limitation. For example,  it remains unclear whether ovarian transposition is effective because you do not have control population without OT, some patients received chemo therapy, and there is no statistical analysis. I think the author should add the limitation in Discussion.  

Reviewer 2 Report

Comments and Suggestions for Authors

comments were modestly applied

Author Response

We further thank the reviewer for his/her comments. We have endeavored to take into account the comments suggested by all the reviewers and to modify the text accordingly. Additional modifications have been made in this latest version.